# Diagnostic and Therapeutic Applications of Exosomes in Cancer with a Special Focus on Head and Neck Squamous Cell Carcinoma (HNSCC)

**DOI:** 10.3390/ijms21124344

**Published:** 2020-06-18

**Authors:** Eliane Ebnoether, Laurent Muller

**Affiliations:** 1Department of Biomedicine, University of Basel, 4031 Basel, Switzerland; Eliane.Ebnoether@usb.ch; 2Department of Otorhinolaryngology, Head and Neck Surgery, University Hospital of Basel, 4051 Basel, Switzerland

**Keywords:** head and neck squamous cell carcinoma (HNSCC), exosomes, cancer, biomarker, diagnostic, therapy, liquid biopsy

## Abstract

Exosomes are nanovesicles part of a recently described intercellular communication system. Their properties seem promising as a biomarker in cancer research, where more sensitive monitoring and therapeutic applications are desperately needed. In the case of head and neck squamous cell carcinoma (HNSCC), overall survival often remains poor, although huge technological advancements in the treatment of this disease have been made. In the following review, diagnostic and therapeutic properties are highlighted and summarised. Impressive first results have been obtained but more research is needed to implement these innovative techniques into daily clinical routines.

## 1. Introduction

After the discovery of exosomes decades ago, they have moved into spotlight for various applications. Their potential as promising biomarkers, especially in cancer, has led to new research approaches making personalised medicine more reachable. In the field of head and neck cancer, biomarkers that simplify diagnosis and treatment are scarce.

Head and neck cancer is the sixth leading cancer by incidence worldwide with a poor five-year overall survival rate of 50% [1]. More than 90% are histologically head and neck squamous cell carcinoma (HNSCC). Although progress has been made and therapies have been enhanced, survival rate has not improved significantly. Eventually a lot of patients die due to metastasis and recurrences.

As diagnosis is tedious, needing a lot of clinical experience as well as resources, and the available therapies are toxic and have a huge impact on quality of life, validated biomarkers would support treatment stratification with improved outcome and reduced toxicity.

In this review, interaction within the immune system, potential diagnostic and therapeutic applications of exosomes with focus on HNSCC are highlighted. Changes of exosomal levels with cancer, under radiation and chemotherapy are discussed and therapeutic options like vaccination, immunomodulation and elimination are studied.

## 2. Intercellular Communication by Exosomes

The first descriptions of exosomes date back more than 30 years. In 1983, a study group discovered the “blebbing” of membranes from transferrin into red blood cells due to maturation of reticulocytes [2]. It was only later that the term exosome was born to describe this “blebbing”.

Exosomes were largely dismissed as means of cellular waste and had fallen into oblivion for many decades. Things changed when their ability for intercellular communication, interaction with the immune system and possible use as a vector of drugs were identified [3]. To date, the exact definition of exosomes has led to a lot of discussions. Different types of vesicles, such as exosomes, extracellular vesicles (EVs) and microvesicles (MVs), have been used to describe similar properties intermixing. Up to now, differences between them and exact functions have not been universally defined [4]. The current opinion describes exosomes as double layered nanovesicles (30–50 nm) originating from the endosomal pathway, thus carrying, for example, tsg101 as a typical marker. They are released by all cell types of the body and are present in extracellular spaces and liquids (e.g., blood, saliva, urine) [5]. They can be directly visualised by electron microscopy (see Figure 1). As seen in Figure 1, differential centrifugation and size-exclusion chromatography, usually yields exosomes of different sizes. Typical surface markers are tetraspanins (CD9, CD63, CD81) and Rab proteins (Ras-related in brain) [6].

Exosomes could be compared to hemerodromes in ancient Greece. Pheidippides is said to be the first marathon runner and covered around 240 km in two days to ask for military support in the fight of Marathon. Exosomes are, like Pheidippides, a messenger system allowing intercellular communication. After this huge run, the hemerodrome Pheidippides is said to have collapsed and died [7]. This ancient story may compare to exosomes that are mostly used up after transmission of the message.

Cancer cells especially are thought to use this communication system avidly. As such, they are known to massively produce exosomes, called tumour-derived exosomes (TEX). During their formation, they acquire lipids, nucleic acids and proteins from the parental cell [8]. Thakur discovered in 2014 the representation of the whole cancer genomic DNA in exosomes [9]. TEX carry double-stranded DNA representing the entire genome and reflecting the mutational status of parental tumour cells. From our own results and that of others, we know that exosomes are able to interact with immune cells and assist in tumour immune escape [10,11]. TEX take part in the development, progression and response to treatment in cancer by alternating intercellular communication.

**Figure 1 ijms-21-04344-f001:**
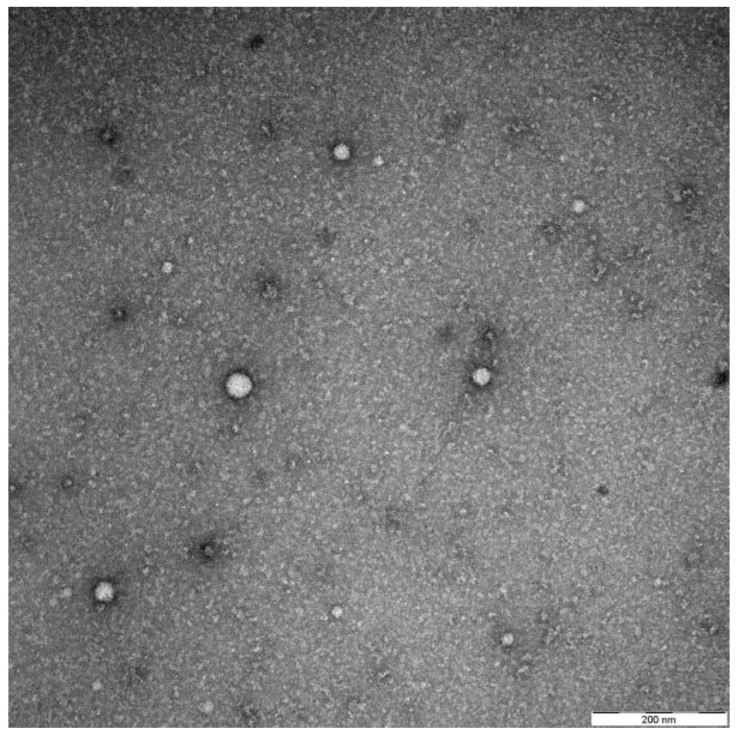
Representative electron microscopy picture by negative staining showing their typical appearance and size range. The exosomes were isolated as described previously from the plasma of a head and neck squamous cell carcinoma (HNSCC) patient [11].

Because of the described properties above, exosomes are ideal candidates to be used as biomarkers and for implementation in new cancer therapies.

## 3. Exosomes in Cancer

Exosomes have gained attention in various cancer types as possible biomarkers. In pancreatic cancer RNA, proteins and DNA in cancer-derived exosomes have been studied. Researchers found 3000 proteins secreted in exosomes derived from pancreatic cancer [12]. Epidermal growth factor receptor (EGFR) seemed to play an important role as binding leads to an increased activity of carcinogenesis signal transduction pathway and binding of associated ligands (EGF, TGF-alpha) is enhanced in most of pancreatic cancer types [13]. Through binding of EGF to EGFR tumour aggressiveness can increase and enhance cell proliferation, migration and probably metastasis [14]. In 2015, Melo et al. described glypican-1 (GPC1) as a specific surface marker for pancreatic cancer. It additionally was able to distinguish normal control subjects from subjects with benign pancreatic lesions [15].

Kahlert et al. assessed possible participation of genomic DNA in cancer-derived exosomes of patients with pancreatic ductal adenocarcinoma (PDAC). Fragments of genomic DNA could be harvested in exosomes and KRAS as well as p53 mutations were detected, giving an opportunity for profiling cancer. These findings underline their use to forecast prognosis and suited management [16].

In breast cancer, exosomal levels of CEA and CA 153 are linked with cancer progression [17,18]. The amount of miRNA in exosomes correlates with malignancy and prognosis [19,20]. A variety of miRNA have been studied and showed an association with breast tumour subtype as well as stage [21].

Concerning the most frequent cancer worldwide, the protein NY-ESO-1 was found to have a significant correlation with survival in lung cancer [22]. In non-small-cell lung cancer miRNA, miR-21 and miR-4257 in exosomes were upregulated in case of recurrence [23]. Decline in miR-51 and miR-373 in lung cancer patients was a factor of poor prognosis [24]. TEX also demonstrated a higher affinity to EGFR resulting in activation of Akt/protein kinase B pathway and overexpression of VEGF and finally augmented tumour vascularity [25]. Ueda et al. described CD91 as specific exosomal marker for lung adenocarcinoma. A panel of exosomal surface marker CD91, CD317 and EGFR was able to differentiate 75% of patients [26,27].

Studies of exosomes in patients with brain tumor showed that exosomes, depending on their cargo, can be used to analyse tumor mutations and predict outcomes of therapy [28]. Analysis of mRNA and total protein expression of patients with glioma after a vaccination trial could show a relation to immunopathological and clinical parameters [29].

Peinado et al. found in 2012 that the size distribution and number of exosomes did not alternate with clinical stage in patients with melanoma, though protein concentration showed a tumour-stage-dependent rise, being highest at the WHO IV stage compared to normal controls and other stages. Protein-poor exosomes in stage IV patients (< 50 μg/mL) showed a survival advantage [30]. Not only could changes in protein levels be detected, but also protein accumulation patterns. Exosomes marked with radioactive iodine-131 in breast-cancer-bearing mice concentrated in the field of the primary tumor as well as in the lung as a premetastastic niche. Moreover, after receiving cancer treatment radioactivity in lung cells diminished and cancer promoting factors declined (VEGFR2, ICAM-1, bFGF, KC, TNF-α, IL-6, IL-12, IL-10 and IL-13) showing a diminution of metastatic competency of the exosomes [31].

These findings implicate similar results. Although a good approach is made, a well-founded, stable biomarker in mentioned carcinomas has not been established up to now.

## 4. Diagnostic Application of Exosomes

### 4.1. Exosomal Blood Levels Correlate With Clinical Disease

To date, for diagnosis of HNSCC and many other cancers, we have to rely on clinical, histopathological and radiological findings. Often a biopsy in the operating room with possible side-effects is inevitable; however, a minimal tumour growth is needed for detection. Exosomes have moved into scope as possible biomarkers, showing early changes in cell properties in real time and being noninvasive.

Skog and his research group showed in 2008 a similar mutational pattern in blood exosomes and glioblastoma cells [32]. In breast cancer, a correlation between rising levels in CEA and CA 153 in exosomes and cancer progression could be shown [9,18,33]. In 2012, Peinado was able to demonstrate an increase in metastatic behaviour in advanced melanoma and exosomal levels in blood correlated well with disease stage [30].

Coming to HNSCC results are very sparse. Studies could show a disease-activity-dependent level of exosomes in HNSCC with advanced-stage tumours exhibiting an elevated level [34]. Other authors showed similar results [35]. Patients had higher exosomal levels in plasma compared to healthy controls. This is the first study describing action of exosomes in this kind of cancer [34]. Finally, the amount of exosomes almost doubled with progression of cancer stage [36].

Aggravated stress situations like inflammation as well as cancer both lead to an increase of exosomes as part of an enhanced intercellular communication, complicating the differentiation between both [37].

Exosomes produced by normal hematopoietic cells mediate antitumour responses and maintain homeostasis, whereas TEX profile changes during the course of disease and treatment. TEX are able to reprogram the bone marrow, cell differentiation migration and organise metastatic tumour spread. These results raised thoughts of a need to subcategorize exosomes as “immune-system-induced” and “tumour-induced”, as functions differ [38]. The Whiteside group separated exosomes in CD3^positive^-fractions (descending from T-lymphocytes) and CD3^negative^-fractions. Hereby, the fraction of immune-derived and tumour-derived exosomes could be studied. In the course of disease, CD3^positive^-exosomes alter their cargo and the ratio of CD3^negative^-exosomes increases, being a marker for cancer progression [38,39]. Clinical response shows an association with found alternations in exosomal cargo.

It has therefore to be considered that TEX show immunosuppressive functions and interference with immune cells, and are used by cancer for tumour escape. A correlation with disease stage and activity could be shown [10,36].

### 4.2. Isolation of Exosomes

Exosomes can be obtained from many body fluids, such as urine, liquor, ascites, saliva or blood of patients [34,40,41]. However, the isolation process is still time-consuming [42,43,44]. Different research groups may use various isolation techniques, which makes their comparison difficult [45]. The international society for extracellular vesicles designed an online course to support with methods [46]. In 2013, a consensus paper for standardisation and isolation was released [47]. Our group uses initial differential centrifugation and size-exclusion chromatography by means of columns. If exosomes will be implicated in daily clinical routine, a standardised, time-saving procedure needs to be developed, which is less prone to errors. Our group is working at the moment on rapid bead-based capturing of exosomes for an easier isolation of exosomes, leading to a drastic time reduction and less vulnerability.

In the future, reproducible and reliable techniques are needed as monitoring tools in the clinical setting. Some starting points have been set but still more results are needed. A huge potential will be, as shown by Hong et al., the increased sensitivity of exosomes compared to standard diagnosis techniques [48,49].

### 4.3. Advantages and Drawbacks of Exosomal Applications

As mentioned above, diagnosis of HNSCC is still strenuous, invasive and linked with great medical expertise. Exosomes, as a potential new diagnostic tool, open up the era of liquid biopsy. As blood collection is a standard procedure in clinical settings, it would be easily applicable to the clinical routine. Further methods like flow cytometry, Western blot or electron microscopy are already established and can be used for continuative examinations. Together with circulating tumor DNA (ct DNA) and circulating tumor cells (CTC), a very sensitive panel for liquid biopsy can be built up, leading to presumably higher sensitivity and earlier diagnosis compared to biopsy or radiography. Initial promising results have been shown in colorectal cancer as well as breast cancer [50,51]. Scholer et al. showed appearance of ct DNA in patients postoperatively after colorectal cancer and detected relapse 9.4 months earlier than CT scan [52]. Compared to ct DNA and CTCs, exosomes seem to be of higher abundance and more stable, which makes them easier to apply into the busy clinical routine.

A persisting main drawback is the lack of standardisation of isolation techniques, which would be important for comparison of results. Besides, platelets in blood seem to secrete exosomes avidly. Already venepuncture and shear forces can lead to release of platelet-derived exosomes and falsification of data [47]. Another not negligible influencing factor is alteration of exosomal levels by infections and other diseases. Owing to the small size of exosomes ranging between 30–150 nm, direct analysis is difficult.

Summing up, the potential of exosomes as diagnostic markers in terms of liquid biopsy is great but prior to clinical application some questions have to be clarified.

### 4.4. Exosome Tropism and Building of a Premetastatic Niche

Biodistribution of TEX shows organ-specific colonisation due to exosomal integrins. Based on this organotropism TEX seem to prepare a premetastatic niche. Clinical results showed that profiles of integrin expression in plasma TEX could be prognostic factors to predict locations for future metastasis. Exosomal integrins are proposed to promote adhesion, trigger signalling pathways and inflammatory responses in target cells leading to educating the organ for expansion of metastatic cells [53]. Garofalo et al. could not support the theory of exosomal integrins responsible for building of the premetastatic niche in their work. They assume a special “receptor/ligand” combination on the exosomes responsible for exosome tropism [54].

Further studies are needed to focus on the mechanism underlying cancer tropism in exosomes as better understanding is important for use of exosome in delivery as theranostic agents and exosomal integrins as a probable diagnostic factor.

### 4.5. Exosomal Alterations with Active Therapy

Changes of plasmatic level of exosomes could not just be shown in cancer but also under active therapy. Patients who underwent surgery had a drastic reduction of plasmatic levels of CD63^+^-exosomes, leading to the assumption, that the tumor mass is responsible for a high number of circulating exosomes. In contrast, exosomes containing Caveolin-1 (CAV-1) increased after surgery with the possible cause directed by inflammation [55]. According to Zorrilla et al. a lower level of exosomes prior and after surgery is linked with increased life expectancy in oral squamous cell carcinoma [55].

Interesting results were found by the group of Lisa Mutschelknaus. As radiation is a common therapy in HNSCC, they discovered a radiation-induced change in the exosomal cargo. It promoted AKT-dependent migration and motility in recipient cancer cells. The AKT pathway is one of the most frequently mutated oncogenic pathways [56,57,58]. The effect was dose-dependent and is an explanation for tumour immune escape and potential driving force for metastatic progression under radiation [59].

Our recent results showed a dramatic decrease of exosomal level seven weeks after primary, combined radio-chemotherapy of HNSCC patients (see Figure 2, results unpublished). This preliminary data demonstrates a positive correlation between clinical response and exosomal blood levels. Although more patients have to be analysed these encouraging results points toward their possible successful use to monitor the therapy.

## 5. Therapeutic Approaches Based On Exosomes

When it comes to therapeutic options, research results are even scarcer. At this point we want to express a warning. Since we are still very early in this research field and the biology behind exosomes is incompletely understood, the application of these new and innovative techniques, although of high research interest, may cause unforeseen harm to patients, and so experiments should be well planned.

### 5.1. Modulation Of Immune System

An imbalance of the immune system might lead to overaction of the inflammatory pathway. Inflammation gets induced through tissue damage or recognition of pathogens. Chronic inflammation has been associated with an increased risk of malignant cell transformation [60,61]. If progression of cancer takes place, different modulations within the immune system are required: apoptosis, proliferation and angiogenesis. The culmination of inflammatory mediators (Stat3, IL-6, TNF-alpha) guides to an immuno-suppression and following progression of cancer. Exosomes are able to participate in this regulation and alteration of immune players with either inhibiting or promoting action.

Exosomes isolated from malignant ascites of ovarian cancer modulated the function of monocytic cells [62]. Production and secretion of proinflammatory cytokines was induced, like interleukin, tumour necrosis factor alpha via TLR2 and 4, leading to an activated nuclear factor as well as signal transducer and activator of transcription (Stat3). The appearance of activated nuclear factor is often observed in cancer cells and results in a more aggressive phenotype with tissue invasion, metastasis and resistance to growth inhibition. [63] Communication between Stat3 and nuclear factor is important for regulation among inflammatory and cancer cells. Apoptosis, angiogenesis and tumour invasion is regulated making resistance towards immune surveillance possible [64].

Apoptosis, as one of the first steps for cancer progression, can be seen through activated T-cells via expression of death ligands (FasL [65] and TRAIL [66]), leading to an evasion of immune surveillance. Concluding modulation with release of FasL-positive exosomes is able to guide immune escape. A regulation of T-cell apoptosis was exemplary shown in human colorectal cancer [67]. Adding up impaired dendritic cell differentiation and suppression of natural killer cells has a similar effect [68,69]. In metastatic ovarian cancer, Yokoi et al. proved that cancer-derived exosomes induced apoptosis after uptake by mesothelial cells perforated the peritoneum resulting in peritoneal dissemination [70].

Concerning proliferation as an important modulation, exosomes derived from thrombin-activated platelets showed an exciting potential for survival, proliferation and chemotaxis of hematopoietic cells [71]. Besides they were capable of activating monocytes and B-cells [72].

Angiogenesis plays an important role during tumour growth, supplying oxygen and nutrition to the cancer and showing effect on an exosomal level [73].

The induction of angiogenesis has been described by various research groups. In 2008, a promotion of angiogenesis could be seen in exosomes secreted by glioblastoma cells [32]. In the same disease, Svensson et al. noted activation of PAR-2 also in vascular endothelial cells. They have arisen through induced angiogenesis after hypoxic treatment [74]. Later, mRNA of exosomes demonstrated proliferation of vascular endothelial cells [75]. An impact while studying renal cancer cell line was seen as CD105-positive exosomes led after affection of vascular endothelial cells to a promotion of growth. CD105-negative cells maintained no effect [76]. Subsequently, promotion of angiogenesis was shown in breast cancer, as miR-210 contained by exosomes seemed to have an impact on it and reversion by suppression of exosomes secretion stated the contrary [77]. Similar findings were reported by Li et al. in liver cancer cells [78].

The migration and invasion of cancer cells are subsequent important factors in metastasis. In colon carcinoma, a transfer of TEX induced a rise of cell proliferation and chemoresistance [79]. In melanoma cells of patients with advanced stages of carcinoma, bone marrow-derived cells were stimulated by exosomes forming a metastatic niche and leading to a prometastatic phenotype [30]. Also, HSP90alpha, as an exosomal surface marker, is able to activate plasminogen to lead to enhanced invasive capacity of tumor cells [80].

Exosomes were studied in relation to T- and NK-cells, as well as macrophages. Exosomes possess the ability of downregulating CD69 expression on CD4^+^ T-cells and interfere with T-cell activation. Exosomes of cancer patients showed even higher immune suppression by decreased levels of CD69 expression. [34] These findings confirm results of other groups [11,81,82,83]. It is suspected that mRNA expression and the translation into inhibitory proteins gets enhanced by TEX.

When looking at CD8^+^ T-cells, coincubation with TEX lets them undergo apoptosis [11].

TEX also exert effects on Treg-cells. The adenosine pathway is used by Treg-cells to operate suppression. A downregulation of mRNA coding for this pathway genes was shown in Treg-cells after coincubation with TEX. Also, TEX appear to activate resting Treg-cells through the increase of CD39 as well as CD73 production. Furthermore, calcium signalling is important for T-cell activation and T-cell-dependent immune responses. Incubation with TEX showed a calcium influx in T-cells, which was stronger than the response induced by dendritic cells [11].

The contribution of B-cells was, to date, not well respected. Results of our experiments could show a suppression of activated B-cells by TEX, which seems to be another way TEX may exert protumour effects (unpublished results).

The exact mechanism of immunomodulation through TEX is still unclear. An important part seems to be the uptake of small noncoding RNA (microRNA, miRNA), which leads to alteration of RNA expression and reprogramming of recipient cells [38]. This was, in particular, shown in immune cells, which take up exosomes as B-cells, monocytes and dendritic cells [82]. miRNA leads to destabilisation of RMA after binding and consecutive reduction of expression of the encoded protein [84]. Some of exosomes-mediated post-transcriptional alternations are associated with horizontal transfer of miRNA and oncogenic proteins from tumour cells to endothelial cells.

Exosomes having miRNA as their cargo are able to shape immune responses and seem responsible for protumorigenic functions in the recipient cells.

TEX signalling happens through receptor binding either on the vesicle surface or an internalisation by the recipient cell takes place. Our work could show that a direct surface interaction of exosomes with T-cells is sufficient to transfer signals. There is no internalisation needed as seen in phagocytic cells [11]. Internalisation and binding of TEX by B-cells and monocytes appeared to be time-dependent, dose-dependent and recipient cell type dependent.

Taken together, TEX take effect in antitumor immune functions and promotion of tumour progression leading to a gain or loss of function in recipient cells and being an option used by tumours as immune escape [30,38,85]. TEX play a dual part, having proinflammatory as well as anti-inflammatory properties. Exosomes of healthy donors did not induce immunosuppression [86].

### 5.2. Loaded Exosomes as Targeted Drug Delivery

Yang et al. showed that unmodified exosomes are able to cross the blood–brain barrier to some extent [87]. Brain exposure can, however, be increased by using certain brain-targeting ligands [88,89].

Exosomes can be modified using various techniques after isolation (exogenous loading): freeze–thaw cycles, incubation, sonication, electroporation and extrusion.

On the contrary, modification can also be made during the formation of exosomes (endogenous loading).

As a breakthrough, Batrakova managed to load exosomes exogenous with the chemotherapeutic agent paclitaxel as the drug-delivery technique. Therapy was, with respect to lung metastasis within mice and in vitro, a lot more successful, having effect even on multiresistant cells and reducing required medication 50 times [90]. Loading drugs into exosomes seems a powerful way of delivering anticancer drugs, but it may lead to increased toxicity, which has to be evaluated carefully.

Another study group loaded glioblastoma-derived exosomes with curcumin (an anti-inflammatory drug) and administered it intranasal to animals with encephalitis. It led to a clinical amelioration with induction of immune tolerance and apoptosis of activated immune cells [91].

Several groups studied the delivery of small interfering RNA (siRNA) into cells using exosomes loaded with electroporation. Some report functional siRNA, others were unsuccessful in proceeding with electroporation [89,92,93,94].

The endogenous approach uses direct transfection of RNA in interest cells, from which exosomes are later derived. Loading with miRNA and siRNA was possible and functional delivery was shown by various groups [94,95,96,97,98]. At the moment, limiting factors of this method are unknown alterations made to the exosome donor cell due to transfection leading to the changed cargo of exosomes. Besides, when conducting transfection, the produced side-products are copurified during isolation, features of exosomes can be changed leading to false-positive effects.

These are promising results for using exosomes as drug delivery vehicles and could be the solution to therapy-resistant cells as well as tumour immune escape.

A huge advantage, the possibility of exosomes to cross the blood–brain barrier, has to be highlighted; this leads to a higher bioavailability regarding the therapeutic use of exosomes if drugs of biologics are administered to them. Additionally, the toxicological effect is selective and targeted, reducing systemic effects compared to conventional free drugs.

However, research is in the fledgling stages and loading of the vesicles can be challenging, with probable alteration of exosomal membrane and function. Another known limitation is well known in cancer research, that once the successful research moves on from animal models to humans, the observed effects may be fading.

### 5.3. Modified Exosomes as Vaccinations

Already two decades ago, dendritic-cell derived exosomes were pulsed with tumour-cell peptides and injected in immune-competent mice. An immunogenic rejection of the tumour was obtained [99]. Due to promising results two clinical trials against melanoma and non-small-cell lung cancer were launched. Regrettably just a small number of patients benefitted from the vaccines.

Some years later, a phase I clinical trial was conducted in China. Extracellular vesicles (EV) from ascites of colorectal cancer patients were taken and combined with granulocyte-macrophage colony-stimulating factor (GM-CSF) to enhance antitumour dendritic cell activity. Patients benefited from GM-CSF combined with EV but not from EV alone [100].

The results show the successful use of exosomes to induce an antitumour immune response. From previous vaccination trials performed in cancer, it is known that the antitumour response is not enough to get rid of the cancer and new studies are being performed to combine immune-modulatory drugs with vaccines to increase their efficiency.

### 5.4. Engineered/Designer Exosomes

The first approaches for delivery drug technologies in the context of loading already existing exosomes have already been described above. This technology bears some problems like probable missing tissue specificity and immunogenic potential.

Alvarez-Erviti et al. engineered exosomes with self-derived immature dendritic cells to reduce immunogenicity and conducted them to express Lamp2b to realise tissue-specificity. Purified exosomes were loaded with siRNA by electroporation fused to a peptide important in the treatment of Alzheimer’s disease and injected intravenously. Results showed specific delivery to the brain with a knockdown of mRNA and peptid by 60% [89].

Rashid et al. highlights further engineering of exosomes and their accumulated future hopes and problems. The perfect cell used to engineer exosomes should be nonimmunogenic, have an unlimited cell passage capacity without mutating, have an ability to produce itself abundantly and be easily modified. They believe to have found these potentials in HEK293-Tcells. Mice with metastatic breast cancer showed slower tumor growth after injection with engineered exosomes combined with a better survival compared to control.

Based on HEK293-Tcells, Kojima et al. even took it a step further and designed exosomes completely out of single components. Therapeutic catalase mRNA as cargo in exosomes showed attenuated neuroinflammation in Parkinson’s disease as a positive results [101].

Designer exosomes are still in their fledgling stages but deliver promising results for further research as drug-delivering technologies.

### 5.5. Exosomes in Immune-Modulatory Therapies

Therapy in HNSCC has been mainly limited to surgery and radio-chemotherapy in the past few years. With growing understanding of the immune system and discovery of CTLA-4, PD-1, its inhibiting functions and suitable antibodies through James Allison and Tasuku Honjo check-point inhibition was found. Antibodies of CTLA-4 showed auspicious results in metastasised melanoma. PD-1 was further studied in HNSCC.

PD-L1 is found on the surface of many cell types being a membrane-bound ligand and upregulated in inflammation or malign situations. After binding to the PD-1 receptor on immune T-cells an activation of T-cells gets suppressed important to keep inflammatory responses regulated, being a checkpoint inhibitor. Tumor cells can adapt the mechanism to evade immune destruction [102]. After the discovery of Honjo, an immune-modulatory treatment with anti-PD-1 treatment was developed, showing some response in metastasised HNSCC. To date, it is the only FDA-approved immune-modulatory therapy with successful results. 

One of the great drawbacks remains the poor response rate of 10–30% with anti-PD-1 treatment [103]. In 2017, isolated exosomes of patients with HNSCC could be shown to carry PD-L1 and PD-1 [36]. PD-L1-carrying exosomes correlate with clinical stage, disease activity and prognosis. Higher numbers of PD-L1^positive^ exosomes leads to poorer prognosis. In contrast, PD-1-containing exosomes did not match with the clinicopathological profile.

Plasma-derived exosomes in HNSCC showed biologically active PD-L1, which managed T-cell dysfunction. Anti-PD-1 was able to reverse this suppression. Maybe exosomal PD-L1, but not PD-1, is clinically relevant, possibly explaining reduced response rate with anti-PD-1 treatment and stratifying clinical responders from nonresponders. [104,105]. Besides, the membrane-bound form PD-L1 was also shown to be freely soluble in blood and other liquids (sPD-L1). Comparing exosome-bound PD-L1 and sPD-L1 levels in HNSCC, the soluble form did not correlate with clinicopathological findings. Results of other studies showed alternating results [106,107].

Exosomal PD-L1 recently moved into the spotlight as a possible therapeutic target, as suppression induces antitumour immunity and memory. Membrane-bound PD-L1 leads to diminished activity of T-cells in lymph nodes. Blocking of exosomal PD-L1 leads to a reversion and suppression of growth of local tumour cells.

Summing up, exosomal PD-L1 is an important regulator of tumour progression with the ability to suppress T-cell activation. Inhibition can lead to antitumour immunity [108].

### 5.6. Blocking/Elimination of Exosomes

A group from Massachusetts observed an interesting finding: incubation of glioma-cell derived exosomes with heparin led to a decreased transfer of exosomes between the parental and recipient cell [109,110]. The blockage happened on the surface and not internally. Further studies concerning the blocking of exosomes or removing them have, to our knowledge, not been conducted.

### 5.7. Challenges to Overcome Application of Exosomes

Exosomes may undergo morphological and functional changes during carcinogenesis, making them elusive for therapeutic applications. In peripheral blood, tumour-associated antigens (TAA) can be found also. Circulating anti-TAA antibodies appear years before clinical diagnosis and are present in all tumor types [111,112,113]. Despite vast research, no specific TAA has been found for HNSCC. Maybe a panel of expression markers will increase sensitivity and specificity.

## 6. Ongoing Clinical Trials with Exosomes in Cancer

In recent years, exosomes have risen to be a trending topic, and a lot of clinical trials have emerged. At the moment, there are 85 active studies listed (www.clinicaltrials.gov, 05 June 2020). Owing to the amount, we omit to list them all. Four studies are listed in the field of HNSCC and exosomes. Further details are mentioned in Figure 3.

The first study investigates grape exosomes as a possible treatment of oral mucositis and examines production of cytokines and immune responses to tumour exosomal antigens and metabolic and molecular markers. The second study examines effects of metformin treatment during radiotherapy on mucositis and change in exosomes and cytokines. The third study is led by Brystol-Myers Squibb. A new biological is tested (BMS986205) for advanced-stage HNSCC. Exosomes are studied in a sidearm and do not belong to the direct treatment. The last study conducted by University of New Mexico analyses a prophylactical swab similar to the cervix PAP-swab and compares it with exosomal levels.

## 7. Conclusions

Since the discovery of exosomes some decades ago, an extended research occurred in this field. Regarding diagnostic possibilities there have been potential approaches, especially within monitoring of HNSCC shown for a follow-up marker (Figure 4). However, for implication in the clinical setting an amendment, stabilisation and standardisation of isolation techniques is needed. Moreover, techniques need to be quick and adequate to present a high sensitivity and specificity in clinical daily routine. There are already ongoing studies.

As in the case of diagnosing recurrences, there is as a complicating factor the hazard of differentiation between recurrence and inflammation (e.g., as result to immunotherapy) as both show increase in exosomal levels and exosomes play a key role in both pathways. This factor aggravates the interpretation of immunotherapy either just being inflammation, tumor necrosis or active recurrence.

In respect of therapeutic options, a great need of action is required. Immunotherapy by PD-1/PD-L1 inhibitors showed already the right direction. A targeted-oriented therapy would be the aim.

Prevailing setbacks in work with exosomes is the not completely understood biology behind them. Therefore, application as targeted drug therapy is delicate as accurate changes and interactions are not known.

Summing up, exosomes seem promising agents to be used for diagnostic or therapeutic purposes in the future, but further studies, including of larger patient cohorts, are needed.

## Figures and Tables

**Figure 2 ijms-21-04344-f002:**
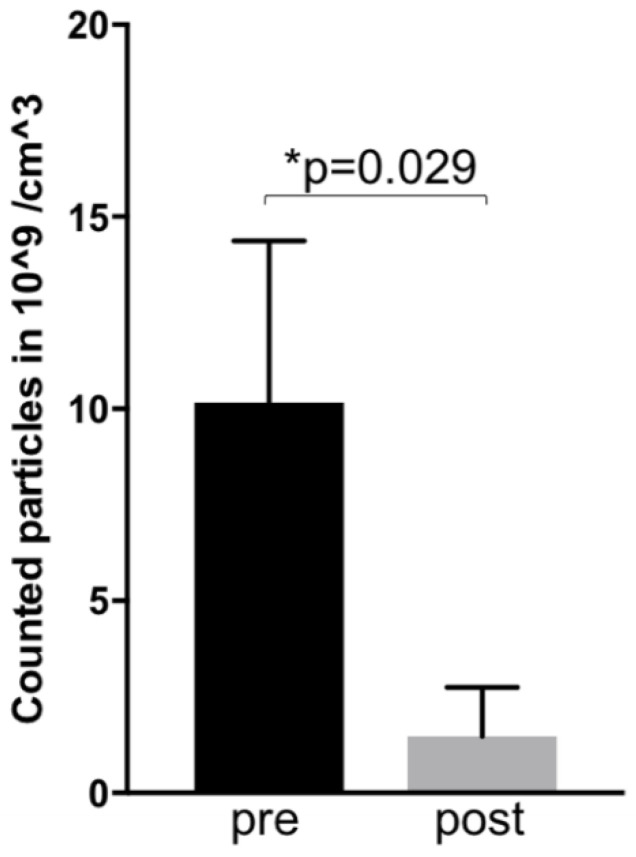
*Preliminary results:* Mean exosomal levels in blood of three patients with active HNSCC before (pre—week 0) and after (post—week 7) primary radio-chemotherapy. The exosomes were isolated as described before [11]. Measurements were performed by Zetaviewer (Particle Metrix, Duesseldorf, Germany). Statistical significance was calculated by a Student *t*-test.

**Figure 3 ijms-21-04344-f003:**
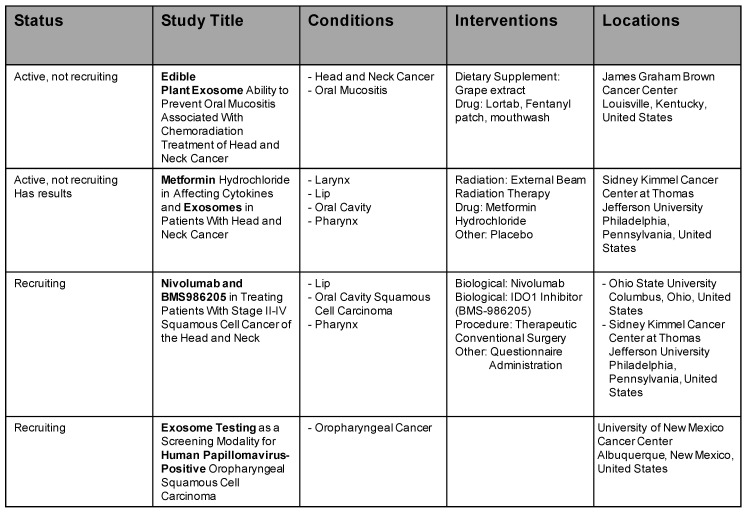
Ongoing clinical trials in cancer involving exosomes in HNSCC.

**Figure 4 ijms-21-04344-f004:**
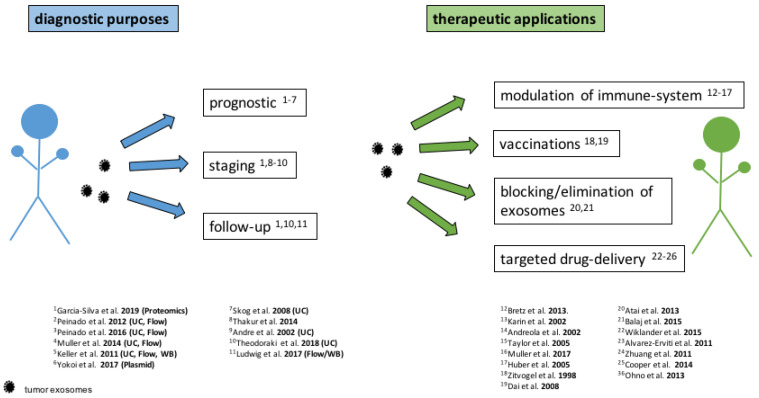
Applications of exosomes in cancer treatment. Isolation technique and method of analysis are written in brackets for the diagnostic purposes. Most research groups used size exclusion chromatography with ultracentrifugation (UC) or flow cytometry (Flow). Some used Western blotting (WB), plasmid transfection (Plasmid) or Fluorescence Activated Cell Sorting (FACS). List of papers consists of a selection.

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
