# Peer review of "Diagnostic and Therapeutic Applications of Exosomes in Cancer with a Special Focus on Head and Neck Squamous Cell Carcinoma (HNSCC)"

_ijms, 2020, doi:10.3390/ijms21124344_

Round 1

Reviewer 1 Report

Comments to the authors:

The manuscript aims to discuss about the potential diagnostic and therapeutic potential of exosomes with a special focus on HNSCC. Although the authors described a few aspects of exosomes as theranostic means, the discussion here is very superficial. I have the following comments –

  1. There are a few fundamental topics are missing in the manuscript, e.g., advantages and disadvantages of exosomal application as a diagnostic and therapeutic marker, challenges to overcome their application, etc.
  2. In figure 1, from the TEM image it is difficult to understand the morphology/appearance of the exosomes. From the scale bar (200nm) all the exosomes are <40-50 nm, which is not possible from a sample. Please check the magnification and the scale bar.
  3. In different sections (section 3, section 5.1), protein contents of exosomes were not discussed while exosomal protens play an indispensable role in cancer progression and immunosuppression. Reference example - https://www.sciencedirect.com/science/article/pii/S154996341930156X.
  4. For section 5.4, it is important to discuss the following references for exosomal PD-L1 –

https://www.nature.com/articles/s41586-018-0392-8

https://www.cell.com/cell/fulltext/S0092-8674(19)30165-5?_returnURL=https%3A%2F%2Flinkinghub.elsevier.com%2Fretrieve%2Fpii%2FS0092867419301655%3Fshowall%3Dtrue

  1. In line 312-313, it is using secreted/releasing appropriate for PD1 and PD-L1 while they are mostly expressed on the cells. Please rephrase the sentence accordingly.
  2. For section 5.2, the authors used “modified” term for loading methods, But they did not discussed engineered exosomes for therapy/diagnostic purposes. Please refer to –

https://www.nature.com/articles/nbt.1807

https://onlinelibrary.wiley.com/doi/full/10.1002/adtp.201900209

Author Response

Response to comment 1:

Thank you for comment. We elaborated the advantages and disadvantages of exosomal application further and added two new passages in the review to further cover these topics (4.3 Advantages and Drawbacks of Exosomal Applications, 5.6 Challenges to Overcome the Application)

Response to comment 2:

We included a better TEM picture.

Response to comment 3:

To better cover the topic of protein contents and the role in cancer progression section 3 has been revised, further literature added and your reference discussed.

Response to comment 4:

Thank you very much for the comment. Exosomal PD-L1 is very important when talking about exosomes in cancer and has to be a topic covered. We added some new references accordingly and discussed them in section 5.5.

Response to comment 5:

The sentence has been rephrased accordingly and made better understandable.

Response to comment 6:

Section 5.2 has been renamed to “loaded exosomes”. To discuss the topic of engineered exosomes further and take the mentioned references into account, a new section has been included. (5.4 Engineered/Designer Exosomes)

Reviewer 2 Report

The Review submitted by Ebnoether and Muller aims to report diagnostic and therapeutic applications of exosomes in cancer.

Nevertheless given the recent findings and the possible use of exosomes to treat cancer, authors should also highlight the importance of exosome-tropism especially considering the use of these vesicles for clinical applications (Hoshino et al Nature 2015; Garofalo et al Theranostics 2019).

Exosomes have been proposed for the delivery of diagnostic agents. This aspect should be also reported

Regarding the therapeutic use of exosomes, they have been proposed to deliver both drugs and biologics with a clear advantage in reducing systemic effects compared to the delivered free drug (not loaded inside exosomes). This aspect should be included as well

Since the review is about the diagnostic and therapeutic applications a table with ongoing clinical trials concerning the use of exosomes in cancer, with a special focus on HNSCC should be reported

Author Response

a) Nevertheless, given the recent findings and the possible use of exosomes to treat cancer, authors should also highlight the importance of exosome-tropism especially considering the use of these vesicles for clinical applications (Hoshino et al Nature 2015; Garofalo et al Theranostics 2019).

Response:

The topic of exosome tropism has been covered in a new section and taken the mentioned references into account. (4.4 Exosome Tropism and Building of a Pre-Metastatic Niche).

b) Exosomes have been proposed for the delivery of diagnostic agents. This aspect should be also reported

Response:

Thank you for your comment. Engineered exosomes is an important topic especially for delivery of therapy. This aspect has been discussed a newly added section (5.4 “Engineered/Designer Exosomes).

c)Regarding the therapeutic use of exosomes, they have been proposed to deliver both drugs and biologics with a clear advantage in reducing systemic effects compared to the delivered free drug (not loaded inside exosomes). This aspect should be included as well

Response:

This aspect has been adapted and included in the passage 5.2 Modified Exosomes as Targeted Drug Delivery.

d)Since the review is about the diagnostic and therapeutic applications a table with ongoing clinical trials concerning the use of exosomes in cancer, with a special focus on HNSCC should be reported

Response:

Exosomes are a “hot topic” and ongoing clinical trial very important. Thank you for mentioning this missing issue. We added the actual trials and an overview table in a new passage (6. Ongoing Clinical Trials).

Round 2

Reviewer 1 Report

Dear authors,

Thank you for editing the manuscript accordingly.

Reviewer 2 Report

I suggest the MS for publication